# Remembering the Past with Today’s Technology: A Scoping Review of Reminiscence-Based Digital Storytelling with Older Adults

**DOI:** 10.3390/bs13120998

**Published:** 2023-12-04

**Authors:** Ling Xu, Noelle L. Fields, M. Christine Highfill, Brooke A. Troutman

**Affiliations:** 1School of Social Work, University of Texas at Arlington, Arlington, TX 76010, USA; noellefields@uta.edu (N.L.F.); christine.highfill@uta.edu (M.C.H.); 2McDermott Library, United States Air Force Academy, Colorado Springs, CO 80840, USA; brooke.troutman@afacademy.af.edu

**Keywords:** digital storytelling, reminiscence, intergenerational engagement, older adults

## Abstract

Reminiscence has been identified as a potentially effective intervention strategy for the mental health of older adults. It has been suggested that reminiscence work and subsequent production of a life storybook (e.g., DST: digital storytelling) is associated with improvements in the well-being of older adults. The specific objectives of this scoping review are to: (1) examine how reminiscence-based DST is conducted/used with older adults, (2) identify whether and how intergenerational engagement is included in this literature, and (3) report on the outcomes identified in this literature, including older adults as well as other participants such as co-creators and viewers of DST. A scoping review following the Joanna Briggs Institute’s methods and the Arksey and O’Malley framework examined studies published in English that included reminiscence with older adults and incorporated digital storytelling. The initial search resulted in 702 articles for review, and following screening, 35 studies were included for full-text review. A total of 10 articles specifically on reminiscence-based DST were identified for final review. Only one study intentionally included intergenerational engagement in its design. DST impacted older adults on their personal meaning and catharsis, social connectedness, cognitive function, and spiritual and emotional well-being. Impacts on reviewers and creators were also reported. Overall, the combination of individual reminiscence work with intergenerational engagement and the use of DST is largely understudied. Additional research is warranted given there is a credible evidence base for these types of interventions.

## 1. Introduction

The use of reminiscence as an intervention or program typically involves a discussion of past activities, events, and experiences using familiar items and objects such as photographs and music [1]. Reminiscence has been identified as a potentially effective intervention strategy for alleviating depression in later life [2,3,4]. Additionally, studies on reminiscence therapy suggest a variety of positive outcomes when used with older adults [5,6], such as promoting social and emotional well-being [7]. Reminiscence work with persons with Alzheimer’s disease and related dementia (ADRD) is also viewed as a psychosocial intervention with a credible evidence base [8]. There is also evidence that individual reminiscence work that involves a life review and subsequent production of a life storybook is associated with improvements in the well-being of persons with ADRD [9,10].

Recent research recommends the production of a digital life storybook [8] as part of reminiscence. One example of a digital life storybook is digital storytelling (DST). DST is a technology that uses a two- to five-minute audio-visual clip combining text, images, music, photographs, voice-over narration, and other audio [11]. As a way to communicate ideas, experiences, beliefs, and topics through the use of technology and multimedia, DST helps storytellers acquire many different skills and literacies as well as learn to create stories using their personal voice and interpretation to be shared with a larger community [12]. DST can be a powerful tool that has been used with the general population and older adults for various purposes. Psychological well-being benefits of using DST for mental health have been reported in a systematic review for the general population [13]. DST with older adults is also well-established in the literature, as it has been successfully used at the end of life [14] and with individuals who have ADRD [15,16] and has benefits of promoting social connectedness [17] and improving sense of self [18].

A few systematic and/or scoping reviews have been conducted on reminiscence with older adults. For example, Yen and Lin [19] systematically reviewed the application and outcomes of reminiscence therapy for older adults in Taiwan. Park and colleagues conducted a systematic review and meta-analysis to identify the effects of reminiscence therapy in people with dementia [20]. Specific reviews have also been conducted for research among older adults with ADRD, such as the effects of reminiscence therapy on people living with dementia and their caregivers [8] and a meta-analysis to investigate the immediate and long-term effects of reminiscence therapy on cognitive functions and depressive symptoms in older persons with dementia [21]. In addition, one systematic review on the use of DST examined the range and extent of the use of digital technologies for facilitating storytelling in older adults and their care partners [16]. Though this review reported 64% of the 34 articles had digital stories related to past events, these stories were not necessarily based on reminiscence.

Reminiscence is different from recalling past events because it is a specific approach that includes a collection of fond memories from the past, and it can be used as a therapeutic technique to bring about feelings of success and confidence [22]. Although research on reminiscence work often endorses the use of a life storybook [20], a preliminary search of systematic and scoping reviews did not yield any literature syntheses on the intersection of reminiscence practices and digital storytelling with older adults. This scoping review specifically focused on studies of reminiscence-based DST.

Moreover, few existing systematic reviews on reminiscence or DST examined how intergenerational engagement or interaction was involved. In this scoping review, intergenerational engagement was broadly characterized as the involvement of a younger generation, whether kinship-related or not, in reminiscence-based DST studies. This involvement encompassed partnering with older adults, participating in reminiscence sharing with older adults, collaborative creation of DST, sharing the DST products, and assessing outcomes for the younger generation and/or the intergenerational relationships. Programs incorporating components of reminiscence with an intergenerational approach are characterized by older persons sharing their life experiences and life lessons to younger generations [23]. Creating DST for older adults can be an innovative approach to teaching and learning in educational settings from elementary schools to universities [24] and, thus, have high potential to involve an intergenerational component. In addition, older adults may encounter challenges or concerns in using technology when creating DST and, thus, may benefit more from DST if the younger generation, who in general may be more confident with the technology, are involved in the process. The positive impact of intergenerational learning experiences has been reported for students’ academic and personal development [24,25,26], as well as having both social and psychological functions for persons with ADRD [27,28]. Two studies that combine reminiscence and an intergenerational program reported positive outcomes including reciprocal relationships, empathy, connection, and confronting ageism [9,24]. However, it is not clear whether these intergenerational reminiscence approaches integrated DST. This scoping review filled this gap by examining intergenerational engagement involvement in the reminiscence-based DST studies.

### Aim and Research Questions of the Present Study

Scoping reviews are a useful tool in the ever-increasing field of evidence synthesis approaches [29]. They are designed to clarify key concepts within the literature, identify types of available evidence, and examine how a particular topic is studied [29]. Therefore, we employed this methodology to address the above-mentioned gaps in the literature. The purposes of this scoping review are to: (1) examine how reminiscence-based DST is conducted/used with older adults, (2) identify whether and how intergenerational engagement is included in this literature, and (3) report on the outcomes identified in this literature, including outcomes of older adults as well as other participants, such as co-creators and viewers of DST.

Overall, in contrast to previous reviews that primarily synthesized the literature on reminiscence [8,19,21] or digital storytelling (DST) [16] among older adults, this review on reminiscence-based DST studies is unique. While it shared similarities with the DST review conducted by Rios and colleagues [16] that focused on DST among older adults, our present review differed in terms of study objectives, inclusion criteria, scope of focus (emphasizing reminiscence-based content), and the incorporation of an intergenerational component. Notably, we also explored outcomes for other participants, such as co-creators and viewers of DST, further contributing to the uniqueness of this review.

## 2. Methods

This review was informed by the JBI approach [30] and the PRISMA extension for Scoping Reviews (PRISMA-ScR) checklist [31] (see Appendix A). It was scaffolded by Colquhoun et al. [32] who synthesized the work of Levac et al. [33] with the seminal work of Arksey and O’Malley [34]. Colquhoun et al. [32] recommended that the scoping review methodology incorporate the frameworks developed by Arksey and O’Malley and expanded by Levac and colleagues, which includes the following: (1) formulation of the research questions, (2) identification of appropriate evidence sources, (3) selection of evidence sources, (4) data extraction and charting, and (5) synthesizing and reporting the results. Before we began the review, we registered an *a priori* protocol for this scoping review at the Center for Open Science. The registration will be under embargo until the present study is published or 2 November 2024, whichever comes first.

The first step of the framework is to develop the research question. We applied Levac et al.’s [33] guidance to determine the question with sufficient breadth to be exploratory yet specific enough to include a concept and target population. We used JBI’s PCC (Population (or participants)/Concept/Context) framework to formulate our questions [30]. Older adults were our population. The concept was an intergenerational component in the context of digital storytelling (See Table 1).

To identify appropriate evidence sources and the means to locate them, we followed Levac et al.’s [33] recommendation to assemble a knowledgeable and experienced team. The interdisciplinary review team consisted of doctoral-level researchers in social work and nursing, a research librarian, a PhD social work student, and a masters-level social work student. All reviewers have contributed to scoping reviews. The team followed Arksey and O’Malley’s [34] recommendation to identify a comprehensive search strategy that was feasible with our time and resource constraints. Our search strategy (see Table 2) was designed by an expert in library science, which is consistent with best practices for systematic literature searches [35,36]. This research librarian on the authorial team has extensive experience in conducting scoping reviews as part of their role as a scholarly librarian. The terms and databases were selected by this librarian after an exhaustive preliminary search, which ruled out databases and terms that did not yield relevant results.

Arksey and O’Malley’s [34] next step is to select the data sources. Scoping reviews are, by nature and design, iterative [33]. To begin, our team identified inclusion and exclusion criteria (See Table 1). To explore the literature explicating reminiscence-based digital storytelling, we applied no date limitation, but required that evidence sources were published in English. We excluded studies that used reminiscence digital storytelling with individuals other than older adults or that examined digital storytelling without reminiscence or reminiscence without digital storytelling. We also excluded other evidence syntheses to avoid confounding our results but conducted a hand screen of their references for possible relevant articles. Using COVIDENCE (www.covidence.org (accessed on 1 December 2023)) systematic review software [37], two authors independently conducted the title and abstract screen and resolved differences through consensus, seeking a third expert opinion when necessary. Full article review was conducted independently by three authors. COVIDENCE automatically included articles with two yes votes (see Figure 1). The audit trail was also documented in Appendix A.

The next stage of Arksey and O’Malley’s [34] conduct of scoping reviews is to chart the data. The team edited the COVIDENCE data extraction template to chart article characteristics (location of study, study population, objective, methodology, findings or results, funding source, affiliation of authors) and variables of interest from each study (digital storytelling element, reminiscence element, intergenerational aspect). The final required stage of conducting a scoping review is collating, summarizing, and reporting the results. Two researchers evaluated the results of COVIDENCE-facilitated extraction and resolved differences through consensus. All of their findings were confirmed by the remainder of the team.

## 3. Results

We identified 702 articles and removed 29 duplicates. An additional 639 articles were excluded as irrelevant during the title and abstract screen. The team conducted a full article review on 35 articles, during which 25 were excluded for the following reasons: did not include a reminiscence feature (*n* = 9), did not include a DST feature (*n* = 8), the population did not include at least one older adult (*n* = 7), and evidence synthesis *(n* = 1). We utilized the remaining sample of 10 articles to answer the research question. See Figure 1 for an inclusion flow chart.

### 3.1. Reminiscence-Based DST with Older Adults

Our first research aim was to examine how reminiscence-based DST was conducted/used with older adults (See Table 3). We identified only 10 studies that met our criteria. Reminiscence-based DST with older adults has been studied across the globe. Three of the studies were conducted in the United States [39,40,41]. Two were conducted in the United Kingdom [10,42]. All other countries were represented by one study each: Australia [43], Canada [44], Korea [45], The Netherlands [46], and Taiwan [2].

Additionally, the included studies employed a variety of research designs. Four of the articles were quantitative studies. Bhar et al. [43] utilized a single-arm trial with an O-X-O design. Moon and Park’s [45] and Elfrink et al.’s [46] studies were randomized control trials. Dang et al., (2021) conducted an observational feasibility trial with an O-X-O design. Four of the studies collected only qualitative data. Chen et al. [2] used interviews and case studies across their two-year study. Phoenix and Griffin [42] conducted focus groups. Harlow et al.’s [40] pilot project collected qualitative data through one-on-one interviews. Loe’s (2013) qualitative study used case reports. Two studies used a mixed methods approach. Hausknecht et al. [44] conducted a mixed-methods study with an O-X-O design and focus groups. Subramaniam and Woods [10] created a mixed methods study using quantitative measures and multiple case studies.

The 10 studies had different purposes or aims. Harlow et al. [40] sought to determine the feasibility of conducting reminiscence-based DST with older adults. Chen et al. [2] sought to determine whether reminiscence-based DST could be a catalyst for social interaction among older adults. Another study investigated the therapeutic efficacy of reminiscence-based DST among cancer patients [39]. Moon and Park’s [45] pilot study aimed to decrease neuropsychiatric symptoms of older women with moderate dementia by using reminiscence and DST. Bhar et al. [43] investigated the change in knowledge and understanding of professional caregivers who watched the reminiscence-based DSTs of older adults in their care. Phoenix and Griffin [42] examined how young athletes’ perception of self-aging changed as a result of viewing reminiscence-based DSTs created by older athletes. Elfrink et al. [46] tested the effectiveness of reminiscence-based DST with a subpopulation of people with dementia and investigated the impact of reminiscence-based DST on the distress and quality of life of primary informal caregivers. Hausknecht et al. [44] explored the benefits of creating reminiscence-based DST for older adults as well as the reactions of the story viewers who attended a public showing of the final products. Loe’s [41] study was based on a 10-week course in which students assisted older adults in creating reminiscence-based DSTs. The study explored the experiences of both the students and their community-dwelling older adult partners. Subramaniam and Woods [10] studied both older adults with dementia in care homes as well as their relatives and care staff to determine the acceptability and efficacy of reminiscence-based DST as compared to conventional life storybooks.

### 3.2. Intergenerational Engagement and Reminiscence-Based DST

The second research aim was to examine whether and how intergenerational engagement was included in this literature. Categorized by the extent of intergenerational engagement, four levels were distinguished (see Table 4): very relevant (*n* = 1), moderately relevant (*n* = 4), minimally relevant (*n* = 3), and not relevant (*n* = 2). Among the ten studies, only the study by Loe [41] was intentionally intergenerational in design and also involved measurements of the younger generation and intergenerational impact (very relevant). In this study, students and older adults participated in a lab component of a course on ageing. Students interviewed their community-dwelling older adult partners in two 1-h sessions. Next, pairs created the reminiscence-based DST together, and at the end of the course, all the DSTs were shared with community members, some of whom were learning lab alumni. This study measured the well-being of both older adults and the student volunteers, as well as the intergenerational impact or differences.

Four identified studies included some form of intergenerational engagement (moderately relevant) where the younger generation (either kinship or not) participated in the reminiscence sharing, DST creation, and sharing of DST products. The Bhar et al. [43] study similarly paired university student volunteers with older adults to create the reminiscence-based DST product. Therefore, intergenerational engagement was presented. However, neither outcomes for the students nor the older adults were measured. The study focused on professional care staff (aged 18–58) who viewed the final products. Results were not grouped according to staff age, so an intergenerational impact could not be ascertained. The Elfrink et al. [46] study utilized trained volunteers to support the older participants with dementia and their caregivers in the creation of the Online Life Story Book. While it is likely that some of the volunteers or caregivers (aged 38–88 years, such as a child, niece, or nephew) were of a younger generation than the older adults, the intergenerational component was not assessed in the study. In Subramaniam and Woods’ work [10], older adults were paired with family members who helped them create their reminiscence-based DSTs. Some of the dyads were intergenerational in nature. For example, some of the younger partners were a nephew, granddaughter, or daughter-in-law. However, others were of the same generation, as in the instance of sisters. The results of the study were not analyzed according to age or generational differences. Similarly, Chen et al. [2] incorporated the participants’ existing social network of family and friends into the study to create and facilitate memory sharing. Relationships and generational differences of each participant and their family members or friends were not reported.

Three studies had no intergenerational engagement in the reminiscence-based DST creation, but the final DST products were shared with the younger generation to some degree (minimally relevant). For example, family members were interviewed in the reminiscence part of the DST project but were not involved in the DST creation [45]. Viewers (family, friends, participants, and community members) watched the final DST products created by the older adults themselves in a classroom environment [44]. Similarly, in Phoenix and Griffin’s study [42], DSTs created by older athletes were shown to athletes of a younger generation. In these three studies, the participants viewing the final DST were from a wide range of age groups. Thus, the intergenerational involvement is likely to have been present to a small degree in the context of these studies, in that only some younger people were exposed to the DSTs of older adults; but the impact of the DST on younger generations is invisible in the final analysis.

Finally, two studies conducted by Dang et al. [39] and Harlow et al. [40] lacked intergenerational engagement in both the creation and sharing of DST, placing them in the category of “not relevant”.

### 3.3. Outcomes of Reminiscence-Based DST with Older Adults

The third research aim was to report on the outcomes identified in the literature on reminiscence-based digital storytelling with older adults (See Table 5). Studies tended to focus either only on the older adults themselves, only on those who viewed or helped produce the DSTs, or both. Additionally, the pilot and exploratory studies by Harlow et al. [40] and Moon and Park [45] demonstrated the feasibility of using DST with older adults.

#### 3.3.1. Impact on Well-Being of Older Adults

Eight of the studies reported the impact of reminiscence-based DST on older adults. Only Bhar et al. [43] and Phoenix and Griffin [42] did not, as their focus was on viewers of DST and not the older adults featured in the reminiscence projects. Among the studies that investigated outcomes of older adults, well-being was measured by all of them but was operationalized in different ways, such as personal meaning and catharsis, social connectedness, cognitive function, and spiritual and emotional well-being.

***Emotional and Spiritual Well-being.*** More than half (*n* = 7) of the studies reported on the emotional and spiritual well-being that participants ascribed to their DST projects [2,10,39,40,41,44,45]. For example, studies reported that DST evoked happy memories [10], provided catharsis [39,40], and facilitated meaning making [44].

Moon and Park [45] found that when compared to the control group (*n* = 19), depression decreased significantly among the DST intervention group (*n* = 27) at the post-intervention (*F* = 6.84, *p* = 0.002) and at the four week follow-up (*F* = 7.62, *p* < 0.001). Subramaniam and Woods [10] had a much smaller sample size (*n* = 6) than Moon and Park [45]. Five of them showed improvement in their depression scores after receiving their digital life storybook. The participants’ quality of life scores also increased after receiving their DST product; however, statistical significance was not reported.

Dang et al.’s study [39] assessed general well-being by the Edmonton Symptom Assessment System (ESAS), a measure specific to cancer patients. Nine of the eleven participants showed improvement or no change, but the improvement was not statistically significant. Improvements were documented at follow-up; however, these were measured in scores for pain, depression, anxiety, or well-being. The report did not indicate whether these were statistically significant. These results must be interpreted with caution since the authors did not disclose the number of participants who improved nor reported age differences among those who did and did not report improvement.

***Social Benefits.*** More than half (*n* = 6) of the studies demonstrated how reminiscence-based DST could facilitate social connectedness in older adults [2,10,40,41,44,45]. In addition to creating current social connections, some participants saw DST as a way to connect to future generations [44]. The digital aspects of the reminiscence product were important, as they preserved the participants’ voices in addition to their photographs for future generations.

Moon and Park [45] assessed the behavioral and emotional expressions and responses of engagement by people with dementia in five areas of psychosocial activity: affective, visual, verbal, behavioral, and social engagement. The Cronbach’s alpha for Moon and Park’s study was 0.74. The mean difference between the traditional reminiscence therapy group (*M* = 0.86, *SD* = 6.01) and the digital reminiscence therapy group (*M* = 3.79, *SD* = 3.82) was statistically significant (*p* = 0.011).

Instead of measuring overall engagement, Subramaniam and Woods [10] examined quality of relationships with caregivers. All the six participants showed improvement on the warmth subscale, and four of the six showed improved scores on the conflict subscale. Six of the six relatives either maintained the maximum scores they reported at baseline or improved in both the warmth and conflict subscales.

***Cognitive Function.*** Four studies reported on cognitive function. Moon and Park [45], Elfrink et al. [46], and Subramaniam and Woods [10] employed quantitative measures. Chen et al., (2013) reported on cognitive function qualitatively.

Moon and Park’s [45] study showed no significant results for cognitive function across groups and three time points. Elfrink et al. [46] found no significant differences in neuropsychiatric symptoms between their control group and intervention group across any of the times (baseline, 3 months, 6 months), conditions, or interactions. The authors posited that significance was not achieved because their samples had very few neuropsychiatric symptoms to begin with and therefore had little room for improvement. Although Subramaniam and Woods [10] showed improvement in autobiographical memory for all six of their participants, they did not provide statistical significance for the measure. Although Chen et al. [2] did not measure cognitive function specifically, qualitative findings indicated that the Story Frame helped close memory gaps.

#### 3.3.2. Impact on Important Others

In some studies, important others participated in the creation of reminiscence-based DST products. Different studies investigated the experiences of important others who viewed the final products. On balance, whether important others were instrumental in the creation of the products or viewed them upon completion, they responded favorably to the DST products and processes.

***Co-Creators.*** Loe [41] evaluated the student participants through an open-ended written evaluation and an email inquiry 1–2 years after the project was finalized. Student volunteers reported meaningful reciprocal relationships, linking biography and history, learning to confront ageism and embrace empathy, and reviewing one’s life and making preparations for the next chapter. Some students reported that the continuing relationships with their older adults filled a grandparent void for them, since they did not live in close proximity to their relatives.

Elfrink et al.’s [46] quantitative data also showed modest positive gains. Informal caregivers in Elfrink et al.’s intervention group were supported by volunteers as they worked with their older adult to create a reminiscence-based DST product. They showed statistically significant reductions in self-rated distress when compared to those in the control group. However, they showed no benefit with regard to general distress, distress related to the older adults’ neuropsychiatric symptoms, time investment, care-related quality of life, general quality of life, and life satisfaction.

***Viewers.*** Five of the ten studies reported on the impact of simply watching a DST. Chen et al. [2], Hausknecht et al. [44], Bhar et al. [43], Phoenix and Griffin [42], and Subramaniam and Woods [10] each evaluated important others who viewed completed reminiscence-based DSTs. In aggregate, the studies suggested that simply watching a DST could confer benefits to family, friends, professional caregivers, and strangers.

In Chen et al.’s [2] work, important others were family and friends chosen by the older adults, while Subramaniam and Woods [10] limited their important others to relatives. Chen et al.’s [2] qualitative analysis of family and friends’ experiences demonstrated positive attitudes toward the DST project and the older adult. Subramaniam and Woods [10] also surveyed relatives who viewed the DST projects. They indicated that the project stimulated memories and elicited good feelings, such as enjoyment.

Hausknecht et al.’s [44] important others included family and friends as well as other participants, alumni of the program, and community members at large. Forty-seven viewers participated in the project. Viewers endorsed their favorite stories and provided a short-answer comment about why they liked a given story. The reminiscence-based DSTs that drew the most favor were those that contained lessons and meaning, were stylistically pleasing, and elicited an emotional response.

Both Bhar et al. [43] and Subramaniam and Woods [10] explored the impact of the DST project on staff members. Bhar et al.’s study quantitatively measured the levels of knowledge and understanding professional caregivers had of older adults who lived in their facility. Their paired samples t-test were statistically higher on the post-test assessment of caregivers’ knowledge of residents. A large majority of respondents indicated that watching the DSTs had been beneficial to them (86.8%, *p* < 0.001) and would improve their caregiving (71.7%, *p* < 0.001). In the qualitative portion of the study, 26 of the 31 quotes expressed positivity toward the study. Subramaniam and Woods found that viewing the DST projects of residents increased professional care givers knowledge of residents, encouraged conversation between staff and residents, and facilitated enjoyment and good feelings.

Not all caregivers were positive about the DSTs. In Bhar et al.’s study [43], staff member quotes that were not positive about watching the DSTs tended to explain that they did not learn new information about the residents. One participant suggested that while they did not benefit, a newer employee might. In the Subramaniam and Woods study [10], negative comments were applied to a specific resident who was unable to operate the DVD player. The care staff and his relative believed that the printed storybook would be a more appropriate method of conveying his life story.

Phoenix and Griffin [42] designed a study in which the creators of the reminiscence-based DST had no interaction with the viewers in any capacity. Phoenix and Griffin curated DSTs of aging athletes, shared them with young athletes, and studied the impact of the DSTs on younger athletes’ attitudes. They found that prior to viewing the DSTs, their sample participants described middle age and old age with decidedly negative words (e.g., “bleak, midlife crisis, decrepit”). After watching the DSTs, participants’ comments were more positive (e.g., “hobbies, enjoy life, possibilities, dignified” (see pp. 8–9)).

### 3.4. Perspectives about the Process and DST Product

Four studies presented older adults’ perspectives about the process of creating and sharing their reminiscence-based DST project. In general, the older adults in Chen et al.’s [2] study enjoyed creating and sharing their reminiscence-based DST. Differences arose, however, where digital features that could alter their photographs were concerned. Some participants enjoyed utilizing sepia filters because the filter gave photographs a sense of nostalgia; others associated the gray tones with death. Subramaniam and Woods’ [10] participants also reported that all the older adults in their study enjoyed creating and having a digital life story.

Unlike Chen et al. [2] and Harlow et al.’s [40] studies, all the participants in Dang et al.’s [39] investigation had cancer. Their DST product was the creation of a digital avatar that they used to tell their life story. All of the patients in Dang et al. rated each 5-point Likert item a 4 or a 5. Items measured whether participants found the experience beneficial (*M* = 4.6, *SD* = 0.5); would participate again (*M* = 4.55, *SD* = 0.5); would recommend the experience (*M* = 4.6, *SD* = 0.5); helped me reflect on the past, present, or future hopes (*M* = 4.5, *SD* = 0.7); and engaged easily with my Avatar (*M* = 4.6, *SD* = 0.5).

The only study that presented dissenting voices among older adults’ opinions was Subramaniam and Woods [10]. One resident adamantly preferred the print version over the digitized, video version of her life story. However, the issue appeared to be access, rather than content. She was frustrated that she could not operate the DVD player without help but “no one wants to show me!” (p. 1270). The resident’s daughter-in-law shared that the older adult preferred the content of the DST because it contained the voice of her granddaughter and the music.

## 4. Discussion

Reminiscence therapy acts as a vehicle for older adults to recall their life events and/or experiences, to explore the meanings, and to influence the behavior of individuals who have these memories [19]. DST is another promising approach that has been used with older adults for various beneficial purposes [16]. The combination of individual reminiscence work with the subsequent production of DST is viewed as a sound intervention with a credible evidence base [10]. The present scoping review identified 10 articles that used reminiscence and DST approaches in helping older adults and the outcomes of reminiscence-based DST studies. Additionally, the scoping review examined whether intergenerational engagement was included in these studies. This review contributes to the knowledge base of reminiscence-based DST for older adults and confirms its potential benefits for older adults as well as others (e.g., viewers, co-creators) in different measurement domains.

The identified 10 studies on reminiscence-based DST for older adults had different study designs, purposes, or aims. Though it is quite common to involve a younger generation to help with the DST creation, only one study included intentionally designed intergenerational engagement. The other studies tended to focus either on the older adults themselves, those who viewed or helped produce the DSTs, or both. Outcomes were identified across several domains: impact of reminiscence-based DST on well-being of older adults, impact of reminiscence-based DST on important others (e.g., co-creators, viewers), and the opinions of older adults about the process of creating reminiscence-based DST and sharing final DST products. Aspects of well-being of older adults included personal meaning and catharsis, social connectedness, cognitive function, and spiritual and emotional well-being.

Overall, this review on reminiscence-based DST studies is different from previous reviews that either summarized the literature on reminiscence [8,19,21] or DST [16] among older adults. This review is similar but also different from the review on DST by Rios and colleagues [16]. Both reviews focused on DST among older adults and found technology(ies) were implemented or deployed in the process and digital media production was created towards the end. This review also confirms Rios’ review that DST was used with different purposes among the older adult participants. Most of the DST studies were conducted with mixed methods, and few used RCT. Therefore, the effectiveness of DST is low or mixed, and thus more RCT studies with quantitative measures are called for in the future. Though similar findings to the review by Rios and colleagues were found, the present review is different in study objectives, inclusion criteria, scope of focus (reminiscence-based DST), and additionally including an intergenerational component and outcomes of other participants such as co-creators and viewers of DST.

### 4.1. Research Recommendations

Although the quality of the evidence is not assessed as part of a scoping review [31], the 10 identified studies can offer directions for future study design and methodology. For example, in this scoping review, two studies utilized a single-arm trial with an O-X-O design [39,43], and only two studies [44,45] used RCTs. Future studies should consider using the RCTs design as it has been recommended as a strong research design for topics in patient care because of the power and reliability of results [47]. Future RCT studies should measure the effectiveness of the reminiscence-based DST interventions that not only include pre-test and post-test but also use follow-up measures after 1–3 months because the objective of reminiscence therapy is sustained improvement over time [19]. In addition, the RCTs by Elfrink et al.’s [46] (*n* = 42) and Moon and Park’s pilot study [45] (*n* = 49) used small sample sizes. Future RCT studies are needed with large sample sizes to determine effectiveness in real-world settings. It is also recommended that future studies should recruit heterogeneous participants with differing backgrounds and cultures in order to test the effectiveness of reminiscence-based DST approaches among diverse populations.

Second, the studies in this review employed quantitative (*n* = 4), qualitative (*n* = 4), and mixed (*n* = 2) methods approaches. In the realm of quantitative research, a multitude of measurement tools and outcome indicators were employed, reflecting diversity in research approaches. However, it is imperative to emphasize the necessity of selecting reliable and valid questionnaires and inventories. The choice of these instruments should align with the characteristics of the participants and the overarching purpose of the study. In addition, some research used qualitative methods such as interviews, focus groups, and case studies, which are beneficial in collecting rich psychological information on participants; however, qualitative rigor (credibility, comparability, dependability, and confirmability) [48] was not described in all of the studies. Though scoping reviews do not assess included studies in terms of rigor or methodological quality [31,49], researchers should include their positionality (e.g., background of the researcher, worldview, potential shared experience with participants) and depending on the method, utilize reflexivity to promote transparency in qualitative research. Third, in terms of the intergenerational engagement in the reminiscence-based DST studies, half of the studies (*n* = 5) included some form of intergenerational engagement. However, only one study was intentionally intergenerational in design. Research suggests that reminiscence combined with an intergenerational approach may yield social and mental health benefits for old and young generations, such as reducing social isolation or loneliness [50], preventing stereotypes or negative attitudes toward the older adults from forming, or refute existing ones [51], and building relationships with older adults [27]. Therefore, future studies may consider adding an intergenerational approach to the research design. Moreover, including trained young adult volunteers to the reminiscence-based DST intervention/programs, as opposed to trained professionals, may potentially offer a more sustainable and cost-effective intervention for implementation in community-based settings.

Lastly, the review examined various processes of reminiscence-based DST approaches. In terms of reminiscence, group and individual reminiscence approaches were both used in the literature. A group reminiscence approach is beneficial for older adults who are more reluctant to interact in social groups and may provide them more opportunities for interactions, performances, feedback, and the safety to ensure that all have equal opportunity to participate [19]. The individual reminiscence approach also has its unique benefits, such as a greater willingness to share life review and better facilitation of the relationship between the participants and the leader. In terms of the DST production, a co-creation approach was mostly used in the literature. This is consistent with research using technology with older adults, especially those with cognitive decline, as well as the DST itself in that it is more important how participants see themselves in the co-creation process than the amount of time older adults spend making the overall product or the technical aspects of it [16]. Future study should consider suitable processes based on the study purpose and the characteristics of the participants.

### 4.2. Implications for Practice

This scoping review of studies on reminiscence-based DST approaches suggest that it is a promising intervention/program for older adults, though further studies are necessary to establish evidence-based protocols and systemic effectiveness for intervention that use both reminiscence and DST. Professionals in gerontology and geriatrics, such as social workers, psychologists, occupational therapists, and nurses, might consider the promising evidence of reminiscence-based DST in promoting the well-being of older adults, such as personal meaning and catharsis, social connectedness, cognitive function, and spiritual and emotional well-being, as well as promoting the well-being of significant others (e.g., co-creators or reviewers of DST), such as general caregiving distress, care-related quality of life, general quality of life, and life satisfaction. In implementing a reminiscence-based DST intervention/program, professionals may consider using a co-creation approach of DST, since this emerged in the scoping review as a common strategy that could be implemented in clinical practice. It is also recommended that professionals might consider integrating an intergenerational component to the reminiscence-based DST intervention/program that utilizes trained volunteer young generations to work with older adults due to its multiple potential benefits to both young and older generations, as well as the new technologies involved in DST, which the younger generation may be familiar in using.

### 4.3. Limitations

Despite our efforts to conduct an exhaustive search of health databases, expand the timeframes of the published studies, and be as inclusive as possible when selecting the target population, we may have missed papers that were not published nor indexed in these databases. Another potential limitation is that we may not have captured relevant studies outside of the inclusion and exclusion criteria. However, we worked with our university scholarly librarian in order to attempt to minimize these limitations. Thirdly, the scoping review of reminiscence-based DST studies faced challenges in precisely differentiating between projects that involve DST with reminiscence and those without. This is attributed to the fact that some studies on DST may incorporate reminiscence practices, albeit without explicit clarification. As a result, the scoping review may have inadvertently overlooked certain studies on this topic. Nevertheless, various members of our research team have diligently reviewed all the pertinent articles on DST, conducting thorough searches and screening their contents. Finally, this review only included articles published in English, which limited inclusion of those written in other languages.

## 5. Conclusions

There is growing interest in using technology in interventions with older adults to address problems such as social isolation and loneliness as well as to promote health and psychosocial well-being in later life. There is also evidence that bolsters support for the use of reminiscence combined with an intergenerational approach to help strengthen solidarity among younger and older adults as well as to reduce ageism. The integration of all three elements (DST, intergenerational, reminiscence) is a promising area of research that warrants further attention. Given the predicted increase in the number of older adults with ADRD, it is also critical to consider that family caregivers and health care providers are faced with Increasing demands to meet the psychosocial needs of this population. The use of DST, reminiscence, and an intergenerational approach may be particularly helpful for persons with memory impairment. Finally, interdisciplinary teams that include a researcher with expertise in DST, such as a librarian, may help inform the design and delivery of a multi-component intervention such as those discussed in this scoping review.

## Figures and Tables

**Figure 1 behavsci-13-00998-f001:**
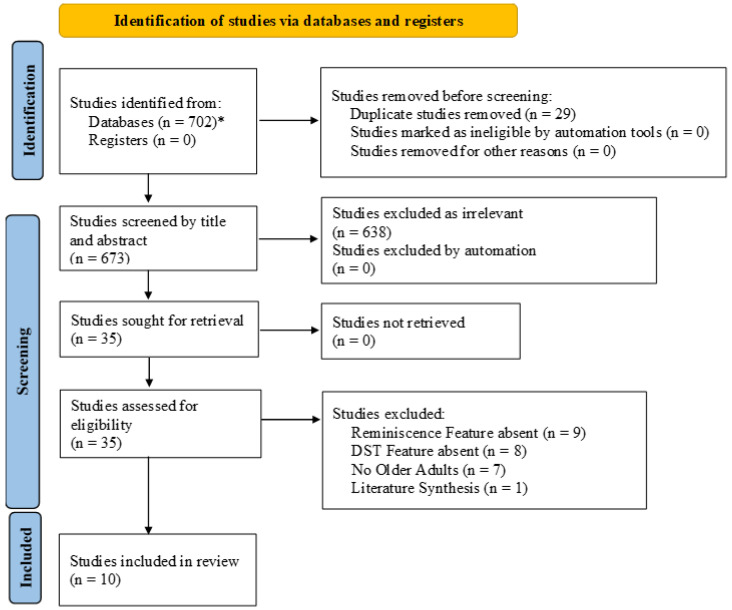
PRISMA flow diagram and screening chart [38]. * Academic Search Complete (*n* = 18); APA PsycInfo (*n* = 13); CINAHL Complete (*n* = 4); ERIC (*n* = 6); MEDLINE (*n* = 10); Social Work Abstracts (*n* = 0); AgeLine (*n* = 0); Web of Science (*n* = 651).

**Table 1 behavsci-13-00998-t001:** Inclusion and exclusion criteria for evidence sources.

Criteria	Definition	Include	Exclude	Related Databases
**Date Range**	Publication dates	Any study published before our last search (2 November 20211)	No study will be excluded based on date of publication	n/a
**Language**	Original or translated	English	All else	n/a
**Method** (Literature Synthesis)	Meta-analysis, scoping review, systematic review, and other evidence synthesis	No evidence synthesis will be included	All evidence synthesis	n/a
**Population** (Older Adults)	Individuals 55 years or over	Studies with at least one participant 55 years or over	Studies with no participants 55 years or over	Academic Search Complete, APA PsycINFO, CINAHL, MEDLINE, Social Work Abstracts, Web of Science
**Concept** (Intergenerational Component)	Two or more generations represented in the study.	Any aspect of the study that includes older adults and young adults	To determine the extent of intergenerational components, no study will be excluded based on this criterion	Academic Search Complete, AgeLine, APA PsycINFO, ERIC, Social Work Abstracts, Web of Science
**Context** (Reminiscence-based Digital Story Telling)	Digital storytelling (e.g., text, video, audio) that includes a reminiscence component (e.g., discussion of the past).	DST that is based on a reminiscence component	All else (e.g., DST without reminiscence; reminiscence without DST)	Academic Search Complete, APA PsycINFO, CINAHL, MEDLINE, Social Work Abstracts, Web of Science

**Table 2 behavsci-13-00998-t002:** Search strategy for evidence sources.

Database	Search String
Academic Search Complete; AgeLine; APA PsycInfo; CINAHL Complete; ERIC; MEDLINE; Social Work Abstracts (Last search 2 November 2021)	(Older adult* OR elder* OR senior* OR geriatric* OR old* people OR aged OR senior citizen* OR 55+ OR aged OR aged 55 OR old* OR old age AND cognitive decline* OR cognitive impairment* OR cognitive dysfunction OR dementia OR Alzheimer’s OR memory loss OR cognitive deficit* OR cognitive function* OR intellect* decline OR intellect* dysfunction OR intellect* loss OR intellect* function OR mental* decline OR mental* loss OR mental* dysfunction) AND (Reminiscence therapy OR reminiscence* OR reminisce* OR life review* OR life story* OR life story work OR life review therapy OR reminisce therapy OR life review treatment* OR reminiscence treatment* OR life story treatment OR remembrance OR remembering OR recollection therapy OR remembrance therapy) AND (digital storytelling OR digital story* OR digital stories OR digital story telling OR DST OR digital narrative* OR digital history OR digital memoir OR digital biography)
Web of Science (Last search 2 November 2021)	Older adult* OR elder* OR senior* OR geriatric* OR old* people OR aged OR senior citizen* OR 65+ OR aged OR aged 65 OR old* OR old age AND cognitive decline* OR cognitive impairment* OR cognitive dysfunction OR dementia OR Alzheimer’s OR memory loss OR cognitive deficit* OR cognitive function* OR intellect* decline OR intellect* dysfunction OR intellect* loss OR intellect* function OR mental* decline OR mental* loss OR mental* dysfunction (All Fields) AND Reminiscence therapy OR reminiscence* OR reminisces* OR life review* OR life story* OR life story work OR life review therapy OR reminisces therapy OR life review treatment* OR reminiscence treatment* OR life story treatment OR remembrance OR remembering OR recollection therapy OR remembrance therapy (All Fields) ANDdigital storytelling OR digital story* OR digital stories OR digital story telling OR DST OR digital narrative* OR digital history OR digital memoir OR digital biography (All Fields)

**Table 3 behavsci-13-00998-t003:** Characteristics of included data sources.

Author (Date)	Location	Study Population *n*/Characteristics	Objective	Design	Funding Sources	Author Affiliation
Bhar et al., (2021) [43]	Australia	Care staff (*n* = 53) male = 14 female = 30 ages 18–58	To examine impact of digital stories of residents on residential care staff	Single-arm Trial OXO	Relationships Australia Victoria	School of Health Sciences Swinburne University of Technology, Australia
Chen et al., (2013) [2]	Taiwan	Study five groups of unspecified composition Minimum age 65 years	To explore whether technology can facilitate reminiscence and social activities	Three-phased exploratory qualitative case study	Centre of Innovation and Synergy for Intelligent Home and Living Technology	Department of Industrial and Commercial Design, National Taiwan University of Science and Technology
Dang et al., (2021) [39]	United States	Advanced cancer patients (*n* = 11) ages 24–54 *(n* = 4) ages 55+ *(n* = 7)	To determine whether digital life review would be enhanced with patient avatars	Observational Feasibility Trial OXO	Massey Cancer Center at Virginia Commonwealth University	Virginia Commonwealth University Health
Elfrink (2021) [46]	The Netherlands	*n* = 42 male = 19 female = 23 ages 49 to 95	Online Life Storybook	Randomized Control Trial	ZonMw, Alzheimer Nederland, and PGGM	University of Twente, Enschede, The Netherlands
Harlow et al., (2003) [40]	United States	*n* = 10 ages over 60	Determine technology applicability in life review	Qualitative Pilot Study	Not disclosed	University of Rhode Island
Hausknect et al., (2019) [44]	Canada	*n* = 80 older adults male = 15 female = 68 ages < 55 to 90; *n* = 47 DST viewers	To assess older adult participants’ benefits of completing a DST course and to determine the opinions of individuals who viewed DST products at a community event	Mixed Methods OXO	Social Sciences and Humanities Research Council of Canada and AGE-WELL NCE Inc.	Simon Fraser University, Burnaby, BC Canada
Loe (2013) [41]	United States	Number of students/older adults not provided. Students were upperclassmen undergraduates. Most were 19–20 years old. Most older adults were in 60s and 70s	To describe the “Digital Life History Project”	Qualitative Case Report	None Disclosed	Colgate University, Hamilton, New York
Moon & Park (2020) [45]	Korea	*n* = 49 all female, over the age of 65. Mean age 84.05 (SD = 6.23)	To evaluate the effect of digital reminiscence therapy as compared with conventional reminiscence therapy	Pilot Study; Randomized Control Trial	Basic Science Research Program through the National Research Foundation funded by the Ministry of Education, Republic of Korea	Department of Nursing Gangneung-Wonjy National University
Phoenix & Griffin (2012) [42]	United Kingdom	*n* = 11 male = 6 female = 5 young adults	To show how older adult athletes stories impact young athletes	Qualitative; focus group study	Funded project, but funder not disclosed	European Centre for the Environment and Human Health
Subramaniam & Woods (2016) [10]	United Kingdom	*n* = 6 male = 2 female = 4	To establish evidence base for acceptability and efficacy of digital life storybooks	Participatory Design; Mixed Methods	None disclosed	Dementia Services Development Centre, Health Psychology Program

**Table 4 behavsci-13-00998-t004:** Intergenerational engagement.

Studies	Reminiscence Sharing	DST Creation	DST Product Sharing	Outcomes Measured	Level of Relevance
Loe (2013) [41]	Students interviewed older adult partners	Co-created DST	Shared with community members	Well-being of young generation and intergenerational relationships	Very relevent
Bhar et al., (2021) [43]	College students paired with older adults	Co-created DST	Shared with professional care staff (aged 18–58)	None	Moderately relevant
Elfrink et al., (2021) [46]	Volunteers (aged 28–60) paired with caregivers (aged 38–88 years) and older adults	Co-created DST	No mention	None	Moderately relevant
Subramaniam & Woods (2016) [10]	Family members (including grandchildren, nieces) were paried with older adults	Co-created DST	Shared among themseleves	None	Moderately relevant
Chen et al., (2013) [2]	Family and friends (no specific age) paired with older adults	Co-created DST	Shared with friends and family members	None	Moderately relevant
Hausknect et al., (2019) [44]	By older adults(no young generation)	Older adults created DST	Shared with viewers (family, friends, participants, and community members) that may be of the young generation	None	Minimally relevant
Moon & Park (2020) [45]	Family memberes were interviewed for older adults’ reminiscence	Nurses and older adults created DST	No mention	None	Minimally relevant
Phoenix & Griffin (2012) [42]	By older athletes (no young generation)	By older athletes	Shared with young athletes	None	Minimally relevant
Dang et al., (2012) [39]	Life interview by research team	By older adults	None specified	None	Not relevant
Halow et al., (2003) [40]	Life interview by research team	By older adults	None specified	None	Not relevant

**Table 5 behavsci-13-00998-t005:** Data extraction chart.

Author (Date)	DST Form	Reminiscence Aspect	Intergenerational Aspect	Intergenerational Relevance to Outcomes	Outcomes
Bhar et al., (2021) [43]	3–4 min digital life stories created for older adults by students	Undergraduate students employed reminiscence techniques to help older adults create the DST	Students helped older adults create DSTs for staff to watch	Moderately relevant	Statistically significant improvement in care staff’s knowledge and understanding regarding residents
Chen et al., (2013) [2]	Electronic picture frame with voice recording of older person’s reflections	“Story Frame” designed to facilitate memory sharing	Older adults were paired with family or friends to create the DST	Moderately relevant	Identified themes: personal collection, sharing, emotional connectedness, reminiscence, oblivion, era atmosphere
Dang et al., (2021) [39]	Avatar-facilitated patient narratives	Semi-structured life review questions guided narratives	None specified	Not relevant	Cancer symptoms improved for 6 of 11 patients. No changes in spiritual well-being. Qualitative findings showed no negative perceptions of DST creation and some indicated catharsis was achieved
Elfrink et al., (2021) [46]	Online life storybook	Volunteers guided participants in life review	Older adults were paired with volunteers or caregivers to create the DST; some dyads were intergenerational	Moderately relevant	Statistically significant reduction in self-rated caregiver distress
Harlow et al., (2003) [40]	Digitized voice recordings facilitated by computers	Interviews to guide participants’ recollections	None specified	Not relevant	Early technology presented problems with talk-to-type but demonstrated potential in facilitating life review
Hausknect et al., (2019) [44]	Older adults created their own DSTs in a class setting	10-week class which taught storytelling techniques and how to effectively share personal history	DST shared with community members	Minimally relevant	Themes from older adult participants: social connectedness through shared experience and story; reminiscence and reflecting on life; and creating a legacy. Viewers identified three characteristics of DST: meaningful stories, well-constructed stories; and stories that invoked a range of emotion
Loe (2013) [41]	3–5 min digital life story	10-week lab in which students and older adults created the DST	Undergraduate students and older adults collaborate to create a digital story of the older adult’s life	Very relevant	Themes identified by older adults and students: meaningful reciprocal relationships; linking biography and history/embracing a “gerontological imagination; learning to confront ageism and embrace empathy; reviewing a life and making preparations for the next chapter
Moon & Park (2020) [45]	Android application	Intervention group received eight sessions of digital reminiscence therapy	Family or friends were interviewed for reminiscence part but were not involved in the DST	Minimally relevant	Statistically decreased depression and increased engagement in intervention group
Phoenix & Griffin (2012) [42]	Digital storytelling product viewed by young adults	Life history interviews; auto-photography	Young athletes viewed DSTs, no intergenerational interaction	Minimally relevant	Students’ negative perceptions of old age changed to positive perceptions after viewing the DST products of older adults
Subramaniam & Woods (2016) [10]	Life story movie	Digitization of conventional life storybook	Older adults were paired with family members, some dyads were intergenerational	Moderately relevant	Positive regard by participants, family, staff; improved or stable depression scores

## Data Availability

Not applicable.

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
