# Peer review of "Remembering the Past with Today’s Technology: A Scoping Review of Reminiscence-Based Digital Storytelling with Older Adults"

_behavsci, 2023, doi:10.3390/bs13120998_

Round 1

Reviewer 1 Report

Comments and Suggestions for Authors

This is a meaningful research study that aligns with the development trend of reminiscence therapy. However, there are several issues that need to be explained.

1. In Table 1, the inclusion criteria for your population are 55 and older, but you wrote 65 and older in your search strategy in Table 2.

2. In Table 3, can you tell me why it is important to collect characteristics about the source of funding?

3. Why is cognitive function categorized under wellbeing in your results section? Please provide relevant references to support its legitimacy.

Comments on the Quality of English Language

There are a few grammar errors that require modification.

Author Response

This is a meaningful research study that aligns with the development trend of reminiscence therapy. However, there are several issues that need to be explained.

  1. In Table 1, the inclusion criteria for your population are 55 and older, but you wrote 65 and older in your search strategy in Table 2.

RESPONSE: Thank you for bringing this error to our attention. We have corrected the age from 65 to 55 in Table 2

  1. In Table 3, can you tell me why it is important to collect characteristics about the source of funding?

RESPONSE: Thank you for this question. The reason to collect characteristics about the source of funding is that the Preferred Reporting Items for Systematic Reviews and Meta-Analyses Scoping Review Extension (PRISMA-ScR) requires funders of included studies to be disclosed when known. Here is the link: http://www.prisma-statement.org/Extensions/ScopingReviews. Funding is item 22.

  1. Why is cognitive function categorized under wellbeing in your results section? Please provide relevant references to support its legitimacy.

RESPONSE: In the realm of psychological health research, the examination of an individual's cognitive functions, or capacity to think, constitutes a pivotal area of study. Cognitive functioning refers to multiple mental abilities, including learning, thinking, reasoning, remembering, problem solving, decision making, and attention (Fisher et al., 2019, p18). For older adults, cognitive health as one important aspect of brain health is important for older adults (NIH National Institute on Aging, NIA).

Fisher, G. G., Chacon, M., & Chaffee, D. S. (2019). Theories of cognitive aging and work. In Work across the lifespan (pp. 17-45). Academic press.

NIH National Institute on Aging. Cognitive health and older adults. https://www.nia.nih.gov/health/cognitive-health-and-older-adults#:~:text=Cognitive%20health%20%E2%80%94%20the%20ability%20to,aspect%20of%20overall%20brain%20health. Accessed on Nov. 14, 2023.

  1. Comments on the Quality of English Language: There are a few grammar errors that require modification.

RESPONSE: We have carefully reviewed the entire text to ensure that no grammar errors are present.

Reviewer 2 Report

Comments and Suggestions for Authors

This is a very clearly written paper, with the methodology explicitly laid out.

The suggestions for improvement are to be less cut and dried in the discussion. The dichotomy about digital story telling with or without reminiscence is not as sharp as claimed in my experience. Being involved in a project of digital storytelling where memories are key, it seems arbitrary what is included. The authors should either better justify the distinction between DST with and without reminiscence, or preferably in my opinion broaden the criteria they use.

I don't object to there only being ten papers being reviewed, but I do not agree that they represent a clearly defined category.

Similarly what constitutes intergenerational is superficially discussed. Is a kinship relationship relevant? Are cultural factors relevant? Again, while I would encourage intergenerational involvement, I don't believe the conclusion is especially justified from the review where the categories are broader. 

Further, the authors claim a broad basis for the people doing the screening of papers. The domains of the screeners include social work and nursing, but not psychology or design, where I am aware relevant research is being undertaken.

The authors should tone down the sharp distinctions.

Author Response

This is a very clearly written paper, with the methodology explicitly laid out.

  1. The suggestions for improvement are to be less cut and dried in the discussion. The dichotomy about digital story telling with or without reminiscence is not as sharp as claimed in my experience. Being involved in a project of digital storytelling where memories are key, it seems arbitrary what is included. The authors should either better justify the distinction between DST with and without reminiscence, or preferably in my opinion broaden the criteria they use.

RESPONSE: As a way to communicate ideas, experiences, beliefs, and topics through the use of technology and multimedia, DST can be conducted based on any content. However, literature suggested that reminiscence-based production of DST is associated with greater improvements in the well-being of older adults. Our study only focused on reminiscence-based DST studies. We revised the title and the relevant texts to make this clearer. In the limitation section, we also acknowledged the challenges of making a clear distinction between DST with and without reminiscence on page 17.

  1. I don't object to there only being ten papers being reviewed, but I do not agree that they represent a clearly defined category.

RESPONSE: We revised the title and manuscript with clearer inclusion criteria: only reminiscence-based DST studies.

  1. Similarly what constitutes intergenerational is superficially discussed. Is a kinship relationship relevant? Are cultural factors relevant? Again, while I would encourage intergenerational involvement, I don't believe the conclusion is especially justified from the review where the categories are broader. 

RESPONSE: In this scoping review, intergenerational engagement was broadly characterized as the involvement of a younger generation, whether kinship-related or not, in reminiscence-based DST studies. This involvement encompassed partnering with older adults, participating in reminiscence sharing with older adults, collaborative creation of DST, sharing the DST products, and assessing outcomes for the younger generation and/or the intergenerational relationships. We added this definition on page 2 and enhanced the clarity of our results section with a new table to closely align with this definition.  

  1. Further, the authors claim a broad basis for the people doing the screening of papers. The domains of the screeners include social work and nursing, but not psychology or design, where I am aware relevant research is being undertaken.

RESPONSE: Although the screeners were from the disciplines of social work and nursing, the articles themselves were derived though systematic searches of databases that represented a variety of academic disciplines beyond social work and nursing, such as gerontology and psychology. In addition, our team also include a research librarian, who has extensive experience in conducting scoping review as part of their role as a scholarly librarian (See page 4).

  1. The authors should tone down the sharp distinctions.

RESPONSE: We revised the implication for practice to make sure that it is clearer.